# A CONTEXTUAL ONLINE LEARNING THEORY OF BROKERAGE

## ABSTRACT

We study the role of *contextual information* in the online learning problem of brokerage between traders. At each round, two traders arrive with secret valuations about an asset they wish to trade. The broker suggests a trading price based on contextual data about the asset. Then, the traders decide to buy or sell depending on whether their valuations are higher or lower than the brokerage price. We assume the market value of traded assets is an unknown linear function of a $d$-dimensional vector representing the contextual information available to the broker. Additionally, at each time step, we model traders' valuations as independent bounded zero-mean perturbations of the asset's current market value, allowing for potentially different unknown distributions across traders and time steps. Consistently with the existing online learning literature, we evaluate the performance of a learning algorithm with the regret with respect to the *gain from trade*. If the noise distributions admit densities bounded by some constant $L$, then, for any time horizon $T$:

- If the agents' valuations are revealed after each interaction, we provide an algorithm achieving $O(Ld\ln T)$ regret, and show a corresponding matching lower bound of $\Omega(Ld\ln T)$.
- If only their willingness to sell or buy at the proposed price is revealed after each interaction, we provide an algorithm achieving $O(\sqrt{LdT \ln T})$ regret, and show that this rate is optimal (up to logarithmic factors), via a lower bound of $\Omega(\sqrt{LdT})$.

To complete the picture, we show that if the bounded density assumption is lifted, then the problem becomes unlearnable, even with full feedback.

## 1 INTRODUCTION

Inspired by a recent stream of literature (Cesa-Bianchi et al., 2021; Azar et al., 2022; Cesa-Bianchi et al., 2024a; 2023; Bolić et al., 2024; Bernasconi et al., 2024), we approach the bilateral trade problem of brokerage between traders through the lens of online learning. When viewed from a regret minimization perspective, bilateral trade has been explored over rounds of seller/buyer interactions with no prior knowledge of their private valuations. As in Bolić et al. (2024), we focus on the case where traders are willing to either buy or sell, depending on whether their valuations for the asset being traded are above or below the brokerage price.

This setting is especially relevant for over-the-counter (OTC) markets. Serving as alternatives to conventional exchanges, OTC markets operate in a decentralized manner and are a vital part of the global financial landscape.[1] In contrast to centralized exchanges, the lack of strict protocols and regulations allows brokers to take on the responsibility of bridging the gap between buyers and sellers, who may not have direct access to one another. In addition to facilitating interactions between parties, brokers leverage their contextual knowledge and market insights to determine appropriate pricing for assets. By examining factors such as supply and demand, market trends, and other asset-specific information, brokers aim to propose prices that reflect the true value of the asset being

---

[1]In the US alone, the value of assets traded in OTC markets exceeded a remarkable 50 trillion USD in 2020, surpassing centralized markets by more than 20 trillion USD (Weill, 2020). This growth has been steadily increasing since 2016 (www.bis.org, 2022).

traded. This price discovery process is a crucial aspect of a broker's role, as it helps ensure efficient transactions by accounting for the unique circumstances surrounding each asset. Additionally, in many OTC markets, as in our setting, traders choose to either buy or sell depending on the contingent market conditions (Sherstyuk et al., 2020). This behavior is observed across a broad range of asset trades, including stocks, derivatives, art, collectibles, precious metals and minerals, energy commodities like gas and oil, and digital currencies (cryptocurrencies), among others (Bolić et al., 2024).

In the existing literature on online learning for bilateral trade, the contextual version of this problem has never been investigated. This case is of significant interest given that the broker often has access to meaningful information about the asset being traded and the surrounding market conditions *before* having to propose a trading price. This information might help the broker to propose more targeted trading prices by inferring the current market value of the corresponding asset, and ignoring it could be extremely costly in terms of missing trading opportunities. We aim to fill this gap in the online learning literature on bilateral trade to guide brokers in these contextual scenarios.

## 1.1 SETTING

In the following, the elements of any Euclidean space are treated as column vectors and, for any real number $x, y$, we denote their minimum by $x \wedge y$ and their maximum by $x \vee y$.

We study the following problem. At each time $t \in \mathbb{N}$,

- Two traders arrive with private valuations $V_t, W_t \in [0, 1]$ about an asset they want to trade.
- The broker observes a context $c_t \in [0, 1]^d$ and proposes a trading price $P_t \in [0, 1]$.
- If the price $P_t$ lies between the lowest valuation $V_t \wedge W_t$ and highest valuation $V_t \vee W_t$ (meaning the trader with the minimum valuation is ready to sell at $P_t$ and the trader with the maximum valuation is eager to buy at $P_t$), the asset is bought by the trader with the highest valuation from the trader with the lowest valuation at the brokerage price $P_t$.
- Some feedback is disclosed.

At any time $t \in \mathbb{N}$, we denote the hidden *marked value* of the asset currently being traded by $m_t \in [0, 1]$. We assume an unknown linear relation exists between the market value $m_t$ for the asset being traded at time $t$ and the corresponding context $c_t$ the broker observes before proposing a trading price. Specifically, we assume that there exists $\phi \in [0, 1]^d$, unknown to the broker, such that, for each $t \in \mathbb{N}$, it holds that $m_t = c_t^\top \phi$. We model the sequence of contexts $c_1, c_2, \ldots$ as a deterministic $[0, 1]^d$-valued sequence (possibly generated in an adversarial manner by someone who knows the broker's algorithm) that is initially unknown but sequentially discovered by the broker. As a consequence, note that the sequence of market values $m_1, m_2, \ldots$ can change arbitrarily (and even adversarially) from one time step to the next. To account for variability due to personal preferences or individual needs, we assume the traders' valuations are zero-mean perturbations of the market values. More precisely, we assume that there exists an independent family of random variables $(\xi_t, \zeta_t)_{t \in \mathbb{N}}$ such that, for each $t \in \mathbb{N}$, it holds that $\mathbb{E}[\xi_t] = 0 = \mathbb{E}[\zeta_t]$ and $V_t = m_t + \xi_t$ and $W_t = m_t + \zeta_t$.[2]

Following the recent stream of bilateral trade literature investigating the interplay between learning and the regularity of the underlying valuation distributions (Cesa-Bianchi et al., 2021; 2023; Bolić et al., 2024), we focus on the case when the traders' valuation distributions admit densities that are uniformly bounded by some constant $L \geq 1$. We note that this assumption is equivalent to the same uniformly bounded density assumption on the distributions of the noise $\xi_1, \zeta_1, \xi_2, \zeta_2, \ldots$. We will later also analyze what happens when the bounded density assumption is lifted.

Consistently with the existing bilateral trade literature, the reward associated with each interaction is the sum of the net utilities of the traders, known as *gain from trade*. Formally, for any $p, v, w \in [0, 1]$, the utility of a price $p$ when the valuations of the traders are $v$ and $w$ is

$$\mathrm{g}(p, v, w) \coloneqq (\underbrace{v \vee w - p}_{\text{buyer's net gain}} + \underbrace{p - v \wedge w}_{\text{seller's net gain}}) \mathbb{I}\{\underbrace{v \wedge w \leq p \leq v \vee w}_{\text{whenever a trade happens}}\} = (v \vee w - v \wedge w)\mathbb{I}\{v \wedge w \leq p \leq v \vee w\}.$$

---

[2]We remark that we are not assuming that the two processes $(\xi_t)_{t \in \mathbb{N}}$ and $(\zeta_t)_{t \in \mathbb{N}}$ are i.i.d., and in fact the distributions of these random variables may change adversarially over time.

The aim of the learner is to minimize the *regret* with respect to the best function of the contexts, defined, for any time horizon $T \in \mathbb{N}$, as

$$R_T \coloneqq \sup_{p^\star : [0,1]^d \to [0,1]} \mathbb{E}\left[ \sum_{t=1}^{T} \Big( \mathrm{GFT}_t\big(p^\star(c_t)\big) - \mathrm{GFT}_t(P_t) \Big) \right],$$

where we let $\mathrm{GFT}_t(p) \coloneqq \mathrm{g}(p, V_t, W_t)$ for all $p \in [0,1]$, and the expectation is taken with respect to the randomness in $(\xi_t, \zeta_t)_{t \in \mathbb{N}}$ and, possibly, the internal randomization used to choose the trading prices $(P_t)_{t \in \mathbb{N}}$.

Finally, we consider the two most studied types of feedback in the bilateral trade literature. Specifically, at each round $t$, only after having posted the price $P_t$, the learner receives either:

- *Full feedback*, i.e., the valuations $V_t$ and $W_t$ of the two current traders are disclosed.
- *Two-bit feedback*, i.e., only the indicator functions $\mathbb{I}\{P_t \leq V_t\}$ and $\mathbb{I}\{P_t \leq W_t\}$ are disclosed.

The information gathered in the full feedback model reflects *direct revelation mechanisms*, where traders disclose their valuations $V_t$ and $W_t$ prior to each round, but the price determined by the mechanism at time $t$ is based solely on the previous valuations $V_1, W_1, \ldots, V_{t-1}, W_{t-1}$. Conversely, the two-bit feedback model reflects *posted price* mechanisms. In this model, traders only indicate their willingness to buy or sell at the posted price, and their valuations $V_t$ and $W_t$ remain undisclosed.

## 1.2 OUR CONTRIBUTIONS

Under the assumption that the traders' valuations are unknown linear functions of $d$-dimensional contexts perturbed by zero-mean noise with time-variable densities bounded by some $L$, and with the goal of designing *simple* and *interpretable* optimal algorithms, we make the following contributions.

1. We prove a structural result (Lemma 1) with two crucial consequences. First, Lemma 1 shows that posting the traders' (unknown) expected valuation as the trading price would maximize the expected gain from trade. Second, it proves that the loss paid by posting a suboptimal price is at most quadratic in the distance from an optimal one.

2. In the full feedback setting, we introduce an algorithm based on ridge regression estimation (Algorithm 1) and, leveraging the previous lemma, we prove its optimality by showing matching $L d \ln T$ regret upper and lower bounds (Theorems 1 and 2).

3. In the two-bit feedback setting, the prices we post directly affect the information we retrieve. We note that this information is so scarce that it is not even enough to reconstruct bandit feedback. We solve this challenging exploration-exploitation dilemma by proposing an algorithm (Algorithm 2) that decides to either explore or exploit adaptively, based on the amount of contextual information gathered so far, and prove its optimality by showing a $\sqrt{L d T \ln T}$ regret upper bound (Theorem 3) and a matching (up to a $\sqrt{\ln T}$) $\sqrt{L d T}$ lower bound (Theorem 4).

4. Finally, we investigate the necessity of the bounded density assumption: by lifting this assumption, we show that the problem becomes unlearnable (Theorem 5).

To the best of our knowledge, our work is the first to analyze a noisy contextual bilateral trade problem (in fact, the first that analyzes a contextual bilateral trade problem in general) and one of only two works on bilateral trade (the other one being Bolić et al. 2024) where the dependence on *all* relevant parameters is tight. As we discuss in Section 1.3, most related works on non-contextual bilateral trade obtain (at best) a matching dependence in the time horizon only, while those on non-parametric noisy contextual pricing/auctions lack matching lower bounds altogether.

## 1.3 RELATED WORKS

Building upon the foundational work of Myerson and Satterthwaite (Myerson & Satterthwaite, 1983), a rich body of research has investigated bilateral trade from a game-theoretic and best-approximation standpoint (Colini-Baldeschi et al., 2016; 2017; Blumrosen & Mizrahi, 2016; Brustle

et al., 2017; Colini-Baldeschi et al., 2020; Babaioff et al., 2020; Dütting et al., 2021; Deng et al., 2022; Kang et al., 2022; Archbold et al., 2023). For an insightful analysis of this literature, see Cesa-Bianchi et al. (2024a).

Our work builds upon the recent research on bilateral trade within online learning settings. Given the close relationship between our and these existing works, we discuss these connections in detail. First, to the best of our knowledge, the existing online learning literature on bilateral trade *never* discussed contextual problems. In Cesa-Bianchi et al. (2021); Azar et al. (2022); Cesa-Bianchi et al. (2024a; 2023; 2024b); Bernasconi et al. (2024), the authors studied non-contextual bilateral trade problems where sellers and buyers have definite roles. Cesa-Bianchi et al. (2021; 2024a) show that the adversarial setting is unlearnable, and hence they focus on the case where sellers' and buyers' valuations form an i.i.d. process. They obtain a $\sqrt{T}$ regret rate in the full-feedback setting. For the two-bit feedback case, they show that the problem is unlearnable in general, but it turns out to be learnable at a tight regret rate of $T^{2/3}$ by assuming that sellers' and buyers' valuations are independent of each other and they admit a uniformly bounded density. Azar et al. (2022) show that learning is achievable in the adversarial case if the weaker $\alpha$-regret objective is considered. Specifically, in the full-feedback case, they obtain a tight 2-regret rate of $\sqrt{T}$. In the two-bit feedback case, they show that learning is impossible in general, but by allowing the learner to use weakly budget-balanced mechanisms, they recover a 2-regret of order $T^{3/4}$, without a matching lower bound. In a different direction, Cesa-Bianchi et al. (2023; 2024b) show that learning is achievable in the adversarial case if the adversary is forced to be *smooth*, i.e., the sellers' and buyers' valuation distributions may change adversarially over time, but these distributions admit uniformly bounded densities. In the full-feedback case, they obtain a tight $\sqrt{T}$ regret rate. In the two-bit feedback case, they show that the problem is still unlearnable, but, by allowing the learner to use weakly budget-balanced mechanisms, they prove a surprisingly sharp $T^{3/4}$ regret rate. Bernasconi et al. (2024) propose the notion of globally budget-balanced mechanisms, a further relaxation of the weakly budget-balanced notion, under which they show that learning is achievable in the adversarial case at a tight regret rate of $\sqrt{T}$ in the full-feedback case, and at a regret rate of $T^{3/4}$ in the two-bit feedback case, without a matching lower bound. We remark that in all the papers we discussed so far, every two-bit feedback upper bound that requires a bounded density assumption lacks a corresponding lower bound with a sharp dependence on this parameter. The closest to our setting is the one proposed in Bolić et al. (2024). There, the authors study the non-contextual version of our trading problem with flexible sellers' and buyers' roles, with the further assumption that the sellers' and buyers' valuations form an i.i.d. sequence. Under the $M$-bounded density assumption, they obtain tight $M \ln T$ and $\sqrt{MT}$ regret rates in the full-feedback and two-bit feedback settings, respectively. If the bounded density assumption is removed, they show that the learning rate degrades to $\sqrt{T}$ in the full-feedback case and the problem turns out to be unlearnable in the two-bit feedback case. We remark that, interestingly, under the bounded density assumption, we are able to achieve the same regret rates in the contextual version of this problem without requiring that traders share the same valuation distribution, while, without the bounded density assumption, the contextual problem is unlearnable even under full-feedback.

Our linear assumption appears commonly in the literature on digital markets, particularly in problems like pricing and auctions. In Cohen et al. (2016; 2020), the authors first address a deterministic setting, then a noisy one with *known* noise distribution where they obtain a regret rate of order $T^{2/3}$ without presenting a lower bound. The deterministic case has also been investigated in Lobel et al. (2017; 2018); Leme & Schneider (2018; 2022); Liu et al. (2021). Notably, the best results currently known only apply to deterministic settings, while, in the case of noisy linear functions, to the best of our knowledge (Xu & Wang, 2021; Badanidiyuru et al., 2023; Fan et al., 2024; Luo et al., 2024; Chen & Gallego, 2021; Javanmard & Nazerzadeh, 2019; Bu et al., 2022; Shah et al., 2019), the only known guarantees are limited to parametric or semi-parametric settings and a clear general picture of the minimax rates is still missing. In contrast, thanks to our Lemma 1, we are able to address the trading problem even when the noise is non-parametric, obtaining optimal rates (matched by corresponding lower bounds) which are significantly faster than the ones known for contextual auctions and pricing.

Another rich related field explored in its many variants (Hanna et al., 2023; Slivkins et al., 2023; Leme et al., 2022; Foster et al., 2021; 2019; Zhou et al., 2019; Kirschner & Krause, 2019; Metevier et al., 2019; Foster & Krishnamurthy, 2018; Kannan et al., 2018; Oh & Iyengar, 2019; Hu et al.,

2020; Neu & Olkhovskaya, 2020; Wei et al., 2020; Krishnamurthy et al., 2020; Luo et al., 2018; Krishnamurthy et al., 2021) is contextual linear bandits. In its standard form, at the beginning of each round, an action set is revealed to the learner, and the assumption is that the reward (which equals the feedback) is a linear function of the action selected from the action set. Instead, in our setting, the market price is a linear function of the context, while the rewards are linked to the price the learner posts by the non-linear gain from trade function. Moreover, in contrast to contextual bandits, in our 2-bit feedback model, the feedback differs from and is not sufficient to compute the reward of the action the learner selects at every round. For these reasons, the techniques appearing in contextual linear bandits do not directly translate to our problem.

## 2 STRUCTURAL RESULTS

We begin by presenting a structural result whose economic interpretation is as follows: even if the broker does not know the traders' valuation distribution, if these valuations can be modeled as zero-mean noisy perturbations with bounded densities of some market value, then the best price to post to maximize the expected gain from trade is precisely the market value. In particular, this generalizes a similar result appearing in Bolić et al. (2024), which holds under the further assumption that the valuations have the exact same distribution. The following result also gives a representation formula for the expected gain from trade, which implies in particular that the cost of posting a suboptimal price is only quadratic in the distance from the market value. This structural result is the key to unraveling the intricacies of the noisy contextual setting, and it is what ultimately allows us to obtain tight regret guarantees in all settings, distinguishing ours from similar contextual pricing works.

**Lemma 1.** *Suppose that $V$ and $W$ are two $[0,1]$-valued independent random variables, with possibly different densities bounded by some constant $L \geq 1$, and such that $\mathbb{E}[V] = \mathbb{E}[W] =: m$. Then, for each $p \in [0,1]$, it holds that*

$$0 \leq \mathbb{E}\big[g(m,V,W) - g(p,V,W)\big] \leq L\,|m-p|^2 \ .$$

*Proof.* We denote by $F$ (resp., $G$) the cumulative distribution function of $V$ (resp., $W$). For each $p \in [0,1]$, from the Decomposition Lemma in (Cesa-Bianchi et al., 2024a, Lemma 1), it holds that

$$\mathbb{E}\big[(W-V)\mathbb{I}\{V \leq p \leq W\}\big] = F(p)\int_p^1 \big(1-G(\lambda)\big)\,\mathrm{d}\lambda + \big(1-G(p)\big)\int_0^p F(\lambda)\,\mathrm{d}\lambda \ ,$$

$$\mathbb{E}\big[(V-W)\mathbb{I}\{W \leq p \leq V\}\big] = G(p)\int_p^1 \big(1-F(\lambda)\big)\,\mathrm{d}\lambda + \big(1-F(p)\big)\int_0^p G(\lambda)\,\mathrm{d}\lambda \ .$$

Hence, for each $p \in [0,1]$,

$$\mathbb{E}\big[(W-V)\mathbb{I}\{V \leq p \leq W\}\big] = F(p)\int_p^1 \big(1-G(\lambda)\big)\,\mathrm{d}\lambda + \big(1-G(p)\big)\int_0^p F(\lambda)\,\mathrm{d}\lambda$$

$$= F(p)\left(m - \int_0^p \big(1-G(\lambda)\big)\,\mathrm{d}\lambda\right) + \int_0^p F(\lambda)\,\mathrm{d}\lambda - G(p)\int_0^p F(\lambda)\,\mathrm{d}\lambda$$

$$= \int_0^p F(\lambda)\,\mathrm{d}\lambda + (m-p)F(p) - pG(p) + G(p)\int_0^p \big(1-F(\lambda)\big)\,\mathrm{d}\lambda + F(p)\int_0^p G(\lambda)\,\mathrm{d}\lambda$$

$$= \int_0^p (F+G)(\lambda)\,\mathrm{d}\lambda + (m-p)(F+G)(p) - G(p)\left(m - \int_0^p \big(1-F(\lambda)\big)\,\mathrm{d}\lambda\right) + (F(p)-1)\int_0^p G(\lambda)\,\mathrm{d}\lambda$$

$$= \int_0^p (F+G)(\lambda)\,\mathrm{d}\lambda + (m-p)(F+G)(p) - \left(G(p)\int_p^1 \big(1-F(\lambda)\big)\,\mathrm{d}\lambda + \big(1-F(p)\big)\int_0^p G(\lambda)\,\mathrm{d}\lambda\right)$$

$$= \int_0^p (F+G)(\lambda)\,\mathrm{d}\lambda + (m-p)(F+G)(p) - \mathbb{E}\big[(V-W)\mathbb{I}\{W \leq p \leq V\}\big] \ .$$

Rearranging, it follows that, for each $p \in [0,1]$,

$$\mathbb{E}\big[g(p,V,W)\big] = \mathbb{E}\big[(W-V)\mathbb{I}\{V \leq p \leq W\}\big] + \mathbb{E}\big[(V-W)\mathbb{I}\{W \leq p \leq V\}\big]$$

$$= \int_0^p (F+G)(\lambda)\,\mathrm{d}\lambda + (m-p)(F+G)(p) \ .$$

Hence, for any $p \in [0, 1]$, it holds that

$$\mathbb{E}\big[\mathrm{g}(m, V, W) - \mathrm{g}(p, V, W)\big] = \int_p^m \big((F + G)(\lambda) - (F + G)(p)\big) \, \mathrm{d}\lambda \geq 0 \, .$$

Finally, since $F$ and $G$ are absolutely continuous with weak derivative bounded by $L$, by the fundamental theorem of calculus (Bass, 2013, Theorem 14.16) it holds that, for $p \in [0, 1]$,

$$\mathbb{E}\big[\mathrm{g}(m, V, W) - \mathrm{g}(p, V, W)\big] = \int_p^m \int_p^\lambda (F' + G')(\vartheta) \, \mathrm{d}\vartheta \, \mathrm{d}\lambda \leq 2L \int_p^m |\lambda - p| \, \mathrm{d}\lambda = L|m - p|^2 \, . \quad \square$$

As a corollary of Lemma 1, we obtain the following result, that upper bounds the regret in terms of the sum of the squared distances between the prices the algorithm posts and the actual market values.

**Corollary 1.** *Consider the setting introduced in Section 1.1. If the valuations admit densities bounded by a constant $L \geq 1$, then, for any time horizon $T \in \mathbb{N}$, we have*

$$R_T = \mathbb{E}\left[\sum_{t=1}^T \big(\mathrm{GFT}_t(c_t^\top \phi) - \mathrm{GFT}_t(P_t)\big)\right] \leq \sum_{t=1}^T 1 \wedge \Big(L\mathbb{E}\big[|P_t - c_t^\top \phi|^2\big]\Big) \, .$$

*Proof.* Given that for each $t \in \mathbb{N}$ and each $p \in [0, 1]$ it holds that $\mathrm{GFT}_t(p) \in [0, 1]$, we have

$$\sup_{p \in [0, 1]} \mathbb{E}\big[\mathrm{GFT}_t(p) - \mathrm{GFT}_t(P_t)\big] \leq 1 \, ,$$

and hence, recalling that $m_t = c_t^\top \phi$ and that $\mathbb{E}[V_t] = m_t = \mathbb{E}[W_t]$, we also have, for each $T \in \mathbb{N}$,

$$R_T = \sup_{p^\star : [0,1]^d \to [0,1]} \sum_{t=1}^T 1 \wedge \Big(\mathbb{E}\big[\mathrm{g}\big(p^\star(c_t), V_t, W_t\big)\big] - \mathbb{E}\big[\mathrm{g}(P_t, V_t, W_t)\big]\Big)$$

$$\overset{(\circ)}{=} \sum_{t=1}^T 1 \wedge \Big(\mathbb{E}\big[\mathrm{g}\big(c_t^\top \phi, V_t, W_t\big)\big] - \mathbb{E}\big[\mathrm{g}(P_t, V_t, W_t)\big]\Big)$$

$$\overset{(*)}{=} \sum_{t=1}^T 1 \wedge \mathbb{E}\Big[\big[\mathbb{E}\big[\mathrm{g}\big(c_t^\top \phi, V_t, W_t\big) - \mathrm{g}(p, V_t, W_t)\big]\big]_{p=P_t}\Big] \overset{(\circ)}{\leq} \sum_{t=1}^T 1 \wedge \Big(L\mathbb{E}\big[|P_t - c_t^\top \phi|^2\big]\Big) \, ,$$

where $(\circ)$ follows from Lemma 1, and $(*)$ from the Freezing Lemma (Cesari & Colomboni, 2021, Lemma 8). $\quad \square$

## 3 FULL FEEDBACK

In this section, we focus on the full feedback setting, corresponding to direct revelation mechanisms. We show that performing ridge regression to obtain an estimate of the unknown vector $\phi$ and using it as a proxy linear function to convert contexts into prices (Algorithm 1) is enough to achieve logarithmic regret. In the following, we denote by $\mathbf{1}_d$ the $d$-dimensional identity matrix.

---
**Algorithm 1:** Ridge Regression Pricing — Full Feedback

---
Observe context $c_1$, post $P_1 \coloneqq 1/2$, and receive feedback $V_1, W_1$;

Let $x_1 \coloneqq [c_1 \mid c_1]$, let $Y_1 \coloneqq [V_1 \mid W_1]$, and compute $\hat{\phi}_1 \coloneqq (x_1 x_1^\top + d^{-1} \mathbf{1}_d)^{-1} x_1 Y_1^\top$;

**for** *time $t = 2, 3, \ldots$* **do**

    Observe context $c_t$, post $P_t \coloneqq c_t^\top \hat{\phi}_{t-1}$, and receive feedback $V_t, W_t$;

    Let $x_t \coloneqq [x_{t-1} \mid c_t \mid c_t]$, $Y_t \coloneqq [Y_{t-1} \mid V_t \mid W_t]$, and compute $\hat{\phi}_t \coloneqq (x_t x_t^\top + d^{-1} \mathbf{1}_d)^{-1} x_t Y_t^\top$;

---

**Theorem 1.** *Consider the full-feedback setting introduced in Section 1.1. If the learner runs Algorithm 1 and the traders' valuations admit a density bounded by $L \geq 1$, then, for any time horizon $T \in \mathbb{N}$, it holds that $R_T \leq 1 + 4Ld\ln T$.*

*Proof.* Recall that $(\xi_t, \zeta_t)_{t \in \mathbb{N}}$ is an independent family of zero mean random variables each of them admitting a density bounded by $L$, that for any $t \in \mathbb{N}$, it holds that $m_t = c_t^\top \phi$, that $m_t + \xi_t = V_t \in [0, 1]$ and that $m_t + \zeta_t = W_t \in [0, 1]$. For any $t \in \mathbb{N}$, simple calculations show that

$$\mathbb{E}\big[|c_{t+1}^\top \hat{\phi}_t - c_{t+1}^\top \phi|^2\big] = \big(\underbrace{\mathbb{E}\big[c_{t+1}^\top \hat{\phi}_t - c_{t+1}^\top \phi\big]}_{\text{bias}}\big)^2 + \underbrace{\text{Var}[c_{t+1}^\top \hat{\phi}_t]}_{\text{variance}}.$$

which is the well-known decomposition of the quadratic error with bias and variance of the estimator $c_{t+1}^\top \hat{\phi}_t$ for the quantity $c_{t+1}^\top \phi$. Noting that, for each $t \in \mathbb{N}$, it holds that $\mathbb{E}[Y_t^\top] = x_t^\top \phi$, we have,

$$\mathbb{E}\big[c_{t+1}^\top \hat{\phi}_t - c_{t+1}^\top \phi\big] = c_{t+1}^\top (x_t x_t^\top + d^{-1} \mathbf{1}_d)^{-1} x_t x_t^\top \phi - c_{t+1}^\top (x_t x_t^\top + d^{-1} \mathbf{1}_d)^{-1} (x_t x_t^\top \phi + d^{-1} \phi)$$

$$= -c_{t+1}^\top (x_t x_t^\top + d^{-1} \mathbf{1}_d)^{-1} d^{-1} \phi =: (\circ) \,,$$

and hence, by the Cauchy-Schwarz inequality applied to the scalar product $(a, b) \mapsto a^\top (x_t x_t^\top + d^{-1} \mathbf{1}_d)^{-1} b$, by the fact that $(x_t x_t^\top + d^{-1} \mathbf{1}_d)^{-1} \preceq d^{-1} \mathbf{1}_d^{-1}$ (where, for any two symmetric matrices $A_1, A_2$, we say that $A_1 \preceq A_2$ if and only if $A_2 - A_1$ is semi-positive definite), and by the fact that $\|\phi\|_2^2 \leq d$, we can control the bias term as follows

$$\big(\mathbb{E}[c_{t+1}^\top \hat{\phi}_t - c_{t+1}^\top \phi]\big)^2 = (\circ)^2 \leq c_{t+1}^\top (x_t x_t^\top + d^{-1} \mathbf{1}_d)^{-1} c_{t+1} \cdot d^{-1} \phi^\top (x_t x_t^\top + d^{-1} \mathbf{1}_d)^{-1} d^{-1} \phi$$

$$\leq c_{t+1}^\top (x_t x_t^\top + d^{-1} \mathbf{1}_d)^{-1} c_{t+1} \cdot d^{-1} \phi^\top (d^{-1} \mathbf{1}_d)^{-1} d^{-1} \phi \leq c_{t+1}^\top (x_t x_t^\top + d^{-1} \mathbf{1}_d)^{-1} c_{t+1}. \quad (1)$$

For each $t \in \mathbb{N}$, letting $\Delta_t$ be the $2t \times 2t$ diagonal matrix with vector of diagonal elements given by $(\text{Var}[V_1], \text{Var}[W_1], \text{Var}[V_2], \text{Var}[W_2], \ldots, \text{Var}[V_t], \text{Var}[W_t])$, we have

$$\text{Var}[c_{t+1}^\top \hat{\phi}_t] = c_{t+1}^\top (x_t x_t^\top + d^{-1} \mathbf{1}_d)^{-1} (x_t \Delta_t x_t^\top)(x_t x_t^\top + d^{-1} \mathbf{1}_d)^{-1} c_{t+1}. \quad (2)$$

Now, for each $t \in \mathbb{N}$, given that $V_1, W_1, \ldots, V_t, W_t$ are $[0, 1]$-valued, we have that $\Delta_t$ is diagonal with diagonal elements less than 1, and hence $x_t \Delta_t x_t^\top \preceq x_t x_t^\top + d^{-1} \mathbf{1}_d$, which yields a control on the variance term as follows,

$$\text{Var}[c_{t+1}^\top \hat{\phi}_t] \leq c_{t+1}^\top (x_t x_t^\top + d^{-1} \mathbf{1}_d)^{-1} (x_t x_t^\top + d^{-1} \mathbf{1}_d)(x_t x_t^\top + d^{-1} \mathbf{1}_d)^{-1} c_{t+1} = c_{t+1}^\top (x_t x_t^\top + d^{-1} \mathbf{1}_d)^{-1} c_{t+1} \,.$$

In the end, for each $t \in \mathbb{N}$, we have

$$\mathbb{E}\big[|c_{t+1}^\top \hat{\phi}_t - c_{t+1}^\top \phi|^2\big] \leq 2 c_{t+1}^\top (x_t x_t^\top + d^{-1} \mathbf{1}_d)^{-1} c_{t+1} = 2 \|c_{t+1}\|_{(x_t x_t^\top + d^{-1} \mathbf{1}_d)^{-1}}^2$$

$$= 2 \|c_{t+1}\|_{(2 \sum_{s=1}^t c_s c_s^\top + d^{-1} \mathbf{1}_d)^{-1}}^2 = \left\|\sqrt{2} c_{t+1}\right\|_{(\sum_{s=1}^t (\sqrt{2} c_s)(\sqrt{2} c_s)^\top + d^{-1} \mathbf{1}_d)^{-1}}^2 \,, \quad (3)$$

where, for any positive definite matrix $A \in \mathbb{R}^{d \times d}$ and each $u \in \mathbb{R}^d$, we have defined $\|u\|_A := \sqrt{u^\top A u}$. Now, for any time horizon $T \in \mathbb{N}$, leveraging Corollary 1, we have that

$$R_T \leq \sum_{t=1}^T 1 \wedge \Big(L \mathbb{E}\big[|P_t - c_t^\top \phi|^2\big]\Big) \leq 1 + \sum_{t=1}^{T-1} 1 \wedge \Big(L \mathbb{E}\big[|c_{t+1}^\top \hat{\phi}_t - c_{t+1}^\top \phi|^2\big]\Big)$$

$$\leq 1 + L \sum_{t=1}^{T-1} 1 \wedge \left\|\sqrt{2} c_{t+1}\right\|_{(\sum_{s=1}^t (\sqrt{2} c_s)(\sqrt{2} c_s)^\top + d^{-1} \mathbf{1}_d)^{-1}}^2 =: (\star) \,.$$

From here, we apply the elliptical potential lemma (Lattimore & Szepesvári, 2020, Lemma 19.4) to obtain that, for any time horizon $T \in \mathbb{N}$,

$$R_T \leq (\star) \leq 1 + 2Ld \ln\left(\frac{dd^{-1} + 2d(T-1)}{dd^{-1}}\right) = 1 + 2Ld \ln\big(1 + 2d(T-1)\big) \leq 1 + 2Ld \ln(2dT) \,.$$

If $d < T/2$, this implies that $R_T \leq 1 + 2Ld \ln(2dT) \leq 1 + 4Ld \ln T$. If, instead, $d \geq T/2$, then, recalling that $L \geq 1$, we obtain once again that $R_T \leq T \leq 1 + 4Ld \ln T$, concluding the proof. $\square$

We conclude this section by stating a matching worst-case $\Omega(Ld \ln T)$ regret lower bound for any algorithm in the full-feedback case, proving the optimality of Algorithm 1.

At a high level, the proof of this result is based on first building a sequence of contexts defined as a common element of the canonical basis of $\mathbb{R}^d$ during each one of $d$ blocks of $T/d$ consecutive time-steps. Then, in each block, an adaptation of the non-contextual full-feedback lower bound construction in (Bolić et al., 2024, Theorem 3) yields a lower bound of order $L \ln(T/d)$. Summing over blocks gives the result. For a full proof of this result, see Appendix A.

**Theorem 2.** *There exist two numerical constants $a, b > 0$ such that, for any $L \geq 2$ and any time horizon $T \geq \max(4, adL^5, 2d)$, there exists a sequence of contexts $c_1, \ldots, c_T \in [0,1]^d$ such that, for any algorithm $\alpha$ for the contextual brokerage problem with full feedback, there exists a vector $\phi \in [0,1]^d$ and two zero-mean independent sequences $(\xi_t)_{t \in [T]}$ and $(\zeta_t)_{t \in [T]}$ independent of each other, such that if we define $V_t \coloneqq c_t^\top \phi + \xi_t$ and $W_t \coloneqq c_t^\top \phi + \zeta_t$, then for each $t \in [T]$ it holds that $c_t^\top \phi \in [0,1]$, $V_t$ and $W_t$ are $[0,1]$-valued random variables with density bounded by $L$, and the regret of $\alpha$ on the sequence of traders' valuations $V_1, W_1, \ldots, V_T, W_T$ satisfies $R_T \geq bLd \ln T$.*

We remark that the previous lower bound holds even for algorithms that have prior knowledge of the sequence of contexts $c_1, c_2, \ldots$ and that Theorem 1 shows that Algorithm 1 matches the optimal $Ld \ln T$ rate even without this *a-priori* knowledge.

## 4 Two-bit Feedback

In this section, we focus on the two-bit feedback setting, corresponding to posted-price mechanisms. We show that a simple deterministic rule that decides to either explore (by posting a price drawn uniformly in $[0,1]$ to gather feedback to reconstruct the cumulative distribution functions of the traders' valuations) or exploit (by posting the scalar product of the context and the current ridge regression estimate of the unknown weight vector $\phi$) based on the amount of information gathered along the various context dimensions (Algorithm 2) is enough to achieve $\widetilde{\mathcal{O}}\big(\sqrt{LdT}\big)$ regret. We recall that $\mathbf{1}_d$ is the $d$-dimensional identity matrix. Also, for any positive definite matrix $A \in \mathbb{R}^{d \times d}$, we define $\|\cdot\|_A : \mathbb{R}^d \to [0, \infty)$, $v \mapsto \sqrt{v^\top A v}$.

---

**Algorithm 2:** Scouting Ridge Regression Pricing — Two-bit Feedback

---

Post $P_1$ uniformly at random in $[0,1]$, and observe $D_1 \coloneqq \mathbb{I}\{P_1 \leq V_1\}$, $E_1 \coloneqq \mathbb{I}\{P_1 \leq W_1\}$;

Let $b_1 \coloneqq 1$, let $x_1 \coloneqq [c_1 \mid c_1]$, let $Y_1 \coloneqq [D_1 \mid E_1]$ and compute $\hat{\phi}_1 \coloneqq (x_1 x_1^\top + d^{-1} \mathbf{1}_d)^{-1} x_1 Y_1^\top$;

**for** *time* $t = 2, 3, \ldots$ **do**

    Observe context $c_t$ and define $b_t \coloneqq \mathbb{I}\left\{\left\|\sqrt{2}c_t\right\|^2_{(x_{t-1} x_{t-1}^\top + d^{-1}\mathbf{1}_d)^{-1}} > \sqrt{\frac{2d \ln(1 + 2d(T-1))}{LT}}\right\}$;

    **if** $b_t = 1$ **then**

        Post $P_t$ uniformly at random in $[0,1]$, and observe $D_t \coloneqq \mathbb{I}\{P_t \leq V_t\}$, $E_t \coloneqq \mathbb{I}\{P_t \leq W_t\}$;

        Let $x_t \coloneqq [x_{t-1} \mid c_t \mid c_t]$, let $Y_t \coloneqq [Y_{t-1} \mid D_t \mid E_t]$ and compute

        $\hat{\phi}_t \coloneqq (x_t x_t^\top + \mathbf{1}_d)^{-1} x_t Y_t^\top$;

    **else** post $P_t = c_t^\top \hat{\phi}_{t-1}$ and let $x_t \coloneqq x_{t-1}$, $Y_t \coloneqq Y_{t-1}$, and $\hat{\phi}_t \coloneqq \hat{\phi}_{t-1}$;

---

**Theorem 3.** *Consider the two-bit feedback setting introduced in Section 1.1. If the learner runs Algorithm 2 and the traders' valuations admit a density bounded by $L \geq 1$, then, for any time horizon $T$ such that $LT \geq 2d \ln\big(1 + 2d(T-1)\big)$, it holds that $R_T \leq 1 + 4\sqrt{LdT \ln T}$.*

*Proof.* Without loss of generality we assume that $T \geq 2$. Note that for any $t \in \mathbb{N}$, if $b_t = 1$, then

$$\mathbb{E}[D_t] = \mathbb{P}[P_t \leq V_t] = \int_0^1 \mathbb{P}[u \leq V_t]\,\mathrm{d}u = \mathbb{E}[V_t] = \mathbb{E}[c_t^\top \phi + \xi_t] = c_t^\top \phi\,,$$

and, analogously, $\mathbb{E}[E_t] = c_t^\top \phi$. It follows that $\mathbb{E}[Y_t^\top] = x_t^\top \phi$, for any $t \in \mathbb{N}$. Now, for any $t \in \mathbb{N}$, using the very same arguments as in the proof of Theorem 1, from the fact that $\mathbb{E}[Y_t^\top] = x_t^\top \phi$ we can deduce an analogous of (1), and, from the fact that the variances of the random variables $D_1, E_1, \ldots, D_t, E_t$ (for the indexes for which they are defined) are less than or equal to 1, we can deduce an analogous of (2). These two results team up to yield a bound analogous to (3): for $t \in \{2, 3, \ldots\}$,

$$\mathbb{E}\big[|c_t^\top \hat{\phi}_{t-1} - c_t^\top \phi|^2\big] \leq 2\,\|c_t\|^2_{(x_{t-1} x_{t-1}^\top + d^{-1}\mathbf{1}_d)^{-1}}\,.$$

Hence, leveraging Corollary 1, for any $T \in \mathbb{N}$, we have that

$$R_T \le \sum_{t=1}^{T} 1 \wedge \left( L\mathbb{E}\left[|P_t - c_t^\top \phi|^2\right] \right) \le \sum_{t=2}^{T} (1-b_t) L\mathbb{E}\left[|c_t^\top \hat{\phi}_{t-1} - c_t^\top \phi|^2\right] + \sum_{t=1}^{T} b_t$$

$$\le L \sum_{t=2}^{T} (1-b_t) \left\| \sqrt{2}c_t \right\|^2_{(x_{t-1}x_{t-1}^\top + d^{-1}\mathbf{1}_d)^{-1}} + \sum_{t=1}^{T} b_t \le \sqrt{2LdT\ln\left(1 + 2d(T-1)\right)} + \sum_{t=1}^{T} b_t \, .$$

Now, given that $LT/\left(2d\ln\left(1 + 2d(T-1)\right)\right) \ge 1$, using the convention $0/0 = 0$,

$$\sum_{t=2}^{T} b_t = \sum_{t=2}^{T} \frac{b_t \left\| \sqrt{2}c_t \right\|^2_{(x_{t-1}x_{t-1}^\top + d^{-1}\mathbf{1}_d)^{-1}}}{\left\| \sqrt{2}c_t \right\|^2_{(x_{t-1}x_{t-1}^\top + d^{-1}\mathbf{1}_d)^{-1}}} \le \sqrt{\frac{LT}{2d\ln(1 + 2d(T-1))}} \sum_{t=2}^{T} 1 \wedge b_t \left\| \sqrt{2}c_t \right\|^2_{(2\sum_{s=1}^{t-1} b_s c_s c_s^\top + d^{-1}\mathbf{1}_d)^{-1}}$$

$$= \sqrt{LT/\left(2d\ln\left(1 + 2d(T-1)\right)\right)} \sum_{t=1}^{T-1} 1 \wedge \left\| b_{t+1}\sqrt{2}c_{t+1} \right\|^2_{\left(\sum_{s=1}^{t}(b_s\sqrt{2}c_s)(b_s\sqrt{2}c_s)^\top + d^{-1}\mathbf{1}_d\right)^{-1}} =: (*).$$

Using the elliptical potential lemma (Lattimore & Szepesvári, 2020, Lemma 19.4), we obtain

$$\sum_{t=1}^{T} b_t \le 1 + (*) \le 1 + \sqrt{LT/\left(2d\ln\left(1 + 2d(T-1)\right)\right) \cdot 2d\ln\left(1 + 2d(T-1)\right)} = 1 + \sqrt{2LdT\ln\left(1 + 2d(T-1)\right)} \, .$$

Hence, if $d < T/2$, this implies that $R_T \le 1 + 2\sqrt{2LdT\ln\left(1 + 2d(T-1)\right)} \le 1 + 4\sqrt{LdT\ln T}$. On the other hand, if $d \ge T/2$, then, since $L \ge 1$, we obtain, again, $R_T \le T \le 1 + 4\sqrt{LdT\ln T}$. $\qquad\square$

We conclude this section by stating a matching (up to logarithmic terms) worst-case $\Omega\left(\sqrt{LdT}\right)$ regret lower bound for any algorithm in the two-bit-feedback case, proving the optimality of Algorithm 2.

At a high level, the proof of this result is based on the same trick (as in the proof of Theorem 2) of choosing contexts equal to vectors of the canonical basis of $\mathbb{R}^d$ in order to obtain $d$ independent 1-dimensional sub-instances. In each block, an adaptation of the non-contextual full-feedback lower bound construction in Bolić et al. (2024, Theorem 5) yields a lower bound of order $\sqrt{LT/d}$. Summing over blocks gives the result. For more details on the proof of this result, see Appendix B.

**Theorem 4.** *There exist two numerical constants $a, b > 0$ such that, for any $L \ge 2$ and any time horizon $T \ge \max(4, adL^3, 2d)$, there exists a sequence of contexts $c_1, \ldots, c_T \in [0,1]^d$ such that, for any algorithm $\alpha$ for the contextual brokerage problem with two-bit feedback, there exists a vector $\phi \in [0,1]^d$ and two zero-mean independent sequences $(\xi_t)_{t\in[T]}$ and $(\zeta_t)_{t\in[T]}$ independent of each other such that, if we define $V_t := c_t^\top \phi + \xi_t$ and $W_t := c_t^\top \phi + \zeta_t$, then for each $t \in [T]$ it holds that $c_t^\top \phi \in [0,1]$, $V_t$ and $W_t$ are $[0,1]$-valued random variables with density bounded by $L$, and the regret of $\alpha$ on the sequence of traders' valuations $V_1, W_1, \ldots, V_T, W_T$ satisfies $R_T \ge b\sqrt{LdT}$.*

We remark that the previous lower bound holds even for algorithms that have prior knowledge of the sequence of contexts $c_1, c_2, \ldots$ and that Theorem 3 shows that Algorithm 2 matches the optimal $\sqrt{LdT}$ rate (up to a $\sqrt{\ln T}$ factor) even without this *a-priori* knowledge.

## 5 BEYOND BOUNDED DENSITIES

In this final section, we investigate the general case where the valuations of the traders are not assumed to have a bounded density, and we show that the problem is, in general, unlearnable.

At a high level, the main reason why the problem becomes unlearnable is that Lemma 1 and its Corollary 1 fail to hold. In fact, the optimal price at time $t$ depends in general not only on the market value $m_t = c_t^\top \phi$, but also on properties of the *time-varying* distributions of the perturbations $\xi_t$ and $\zeta_t$, which essentially turns our problem into a fully-adversarial one where we strive to compete against time-varying policies. For a full proof of the following theorem, see Appendix C.

**Theorem 5.** *There exists a sequence of contexts $c_1, c_2, \cdots \in [0,1]^d$ and a vector $\phi \in [0,1]^d$, such that for any algorithm $\alpha$ for the contextual brokerage problem under full feedback, there exists an*

*independent sequence of zero mean random variables $\xi_1, \zeta_1, \xi_2, \zeta_2, \ldots$, such that if the valuations of the traders at time $t$ are $V_t = c_t^\top \phi + \xi_t$ and $W_t = c_t^\top \phi + \zeta_t$, then $c_t^\top \phi \in [0, 1]$, $V_t, W_t$ are $[0, 1]$-valued random variables, and the regret of $\alpha$ on the sequence of traders' valuations $V_1, W_1, \ldots, V_T, W_T$ satisfies $R_T = \Omega(T)$.*

We remark that the previous unlearnability result holds even for algorithms that have prior knowledge of the sequence of contexts $c_1, c_2, \ldots$ and, strikingly, of the vector $\phi$.

## 6 CONCLUSIONS

Motivated by the real-life *desideratum* to exploit prior information on the traded assets, we investigated the noisy linear contextual online learning problem of brokerage between traders without predetermined seller/buyer roles. We provided a complete picture with tight regret bounds in all the proposed settings, i.e., under full and two-bit feedback, and with or without regularity assumptions on the noise distributions, achieving tightness (up to $\log$ terms) in all relevant parameters.

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

## A  PROOF OF THEOREM 2

Without loss of generality, we assume that $d$ divides $T$. In fact, if we prove the theorem for this case, then, by leveraging that $T \geq 2d$ and $T \geq 4$, the general case follows from

$$R_T \geq bLd\ln\big(\lfloor T/d \rfloor d\big) \geq \frac{b}{2}Ld\ln T \ .$$

Let $n := T/d$. Let $e_1, \ldots, e_d$ be the canonical basis of $\mathbb{R}^d$. Define, for all $i \in [d]$ and $j \in [n]$, the context $c_{j+(i-1)n} := e_i$. We assume that these contexts are known to the learner in advance and, therefore, we can restrict the proof to deterministic algorithms without any loss of generality.

Let $L \geq 2$, $J_L := \left[\frac{1}{2} - \frac{1}{14L}, \frac{1}{2} + \frac{1}{14L}\right]$, $f := \mathbb{I}_{[0,\frac{3}{7}]} + L\mathbb{I}_{J_L} + \mathbb{I}_{[\frac{4}{7},1]}$, and, for any $\varepsilon \in [-1,1]$, $g_\varepsilon := -\varepsilon\mathbb{I}_{[\frac{1}{7},\frac{3}{14}]} + \varepsilon\mathbb{I}_{(\frac{3}{14},\frac{2}{7}]}$ and $f_\varepsilon := f + g_\varepsilon$. For any $\varepsilon \in [-1,1]$, note that $0 \leq f_\varepsilon \leq L$ and $\int_0^1 f_\varepsilon(x)\,\mathrm{d}x = 1$, hence $f_\varepsilon$ is a valid density on $[0,1]$ bounded by $L$. We will denote the corresponding probability measure by $\nu_\varepsilon$, set $\bar\nu_\varepsilon := \int_{[0,1]} x\,\mathrm{d}\nu_\varepsilon(x)$, and notice that direct computations show that $\bar\nu_\varepsilon = \frac{1}{2} + \frac{\varepsilon}{196}$. Consider for each $q \in [0,1]$, an i.i.d. sequence $(B_{q,t})_{t\in\mathbb{N}}$ of Bernoulli random variables of parameter $q$, an i.i.d. sequence $(\tilde B_t)_{t\in\mathbb{N}}$ of Bernoulli random variables of parameter $1/7$, an i.i.d. sequence $(U_t)_{t\in\mathbb{N}}$ of uniform random variables on $[0,1]$, and uniform random variables $E_1, \ldots, E_d$ on $[-\bar\varepsilon_L, \bar\varepsilon_L]$, where $\bar\varepsilon_L := \frac{7}{L}$, such that $\big((B_{q,t})_{t\in\mathbb{N},q\in[0,1]}, (\tilde B_t)_{t\in\mathbb{N}}, (U_t)_{t\in\mathbb{N}}, E_1, \ldots, E_d\big)$ is an independent family. Let $\varphi\colon[0,1] \to [0,1]$ be such that, if $U$ is a uniform random variable on $[0,1]$, then the distribution of $\varphi(U)$ has density $\frac{7}{6} \cdot f \cdot \mathbb{I}_{[0,1]\setminus[1/7,2/7]}$ (which exists by the Skorokhod representation theorem (Williams, 1991, Section 17.3)). For each $\varepsilon \in [-1,1]$ and $t \in \mathbb{N}$, define

$$G_{\varepsilon,t} := \left(\frac{2+U_t}{14}(1 - B_{\frac{1+\varepsilon}{2},t}) + \frac{3+U_t}{14}B_{\frac{1+\varepsilon}{2},t}\right)\tilde B_t + \varphi(U_t)(1 - \tilde B_t) \ , \tag{4}$$

$V_{\varepsilon,t} := G_{\varepsilon,2t-1}$, $W_{\varepsilon,t} := G_{\varepsilon,2t}$, $\xi_{\varepsilon,t} := V_{\varepsilon,t} - \bar\nu_\varepsilon$, and $\zeta_{\varepsilon,t} := W_{\varepsilon,t} - \bar\nu_\varepsilon$. In the following, if $a_1, \ldots, a_d$ is a sequence of elements, we will use the notation $a_{1:d}$ as a shorthand for $(a_1, \ldots, a_d)$. For each $\varepsilon_1, \ldots, \varepsilon_d \in [-1,1]$, each $i \in [d]$, and each $j \in [n]$, define the random variables

$\xi_{j+(i-1)n}^{\varepsilon_{1:d}} \coloneqq \xi_{\varepsilon_i, j+(i-1)n}$ and $\zeta_{j+(i-1)n}^{\varepsilon_{1:d}} \coloneqq \zeta_{\varepsilon_i, j+(i-1)n}$. The family $\left(\xi_t^{\varepsilon_{1:d}}, \zeta_t^{\varepsilon_{1:d}}\right)_{t\in[T], \varepsilon_{1:d}\in[-1,1]^d}$ is an independent family, independent of $(E_1, \ldots, E_d)$, and for each $i \in [d]$ and each $j \in [n]$ it can be checked that the two random variables $\xi_{j+(i-1)n}^{\varepsilon_{1:d}}, \zeta_{j+(i-1)n}^{\varepsilon_{1:d}}$ are zero mean with common distribution given by $\nu_{\varepsilon_i}$. For each $\varepsilon_1, \ldots, \varepsilon_d \in [-1,1]$, let $\phi_{\varepsilon_{1:d}} \coloneqq (\bar{\nu}_{\varepsilon_1}, \ldots, \bar{\nu}_{\varepsilon_d})$, and for each $i \in [d]$ and $j \in [n]$, let $V_{j+(i-1)n}^{\varepsilon_{1:d}} \coloneqq c_{j+(i-1)n}^\top \phi_{\varepsilon_{1:d}} + \xi_{j+(i-1)n}^{\varepsilon_{1:d}}$ and $W_{j+(i-1)n}^{\varepsilon_{1:d}} \coloneqq c_{j+(i-1)n}^\top \phi_{\varepsilon_{1:d}} + \zeta_{j+(i-1)n}^{\varepsilon_{1:d}}$. Note that these last two random variables are $[0,1]$-valued zero-mean perturbations of $c_{j+(i-1)n}^\top \phi_{\varepsilon_{1:d}}$ with shared density given by $f_{\varepsilon_i}$, and hence bounded by $L$.

We will show that any algorithm has to suffer the regret inequality in the statement of the theorem if the sequence of evaluations is $V_1^{\varepsilon_{1:d}}, W_1^{\varepsilon_{1:d}}, \ldots, V_T^{\varepsilon_{1:d}}, W_T^{\varepsilon_{1:d}}$, for some $\varepsilon_1, \ldots, \varepsilon_d \in [0,1]$.

Before doing that, we first need the following. For any $\varepsilon_1, \ldots, \varepsilon_d \in [-1,1]$, $p \in [0,1]$, and $t \in [T]$ let $\mathrm{GFT}_t^{\varepsilon_{1:d}}(p) \coloneqq \mathrm{g}(p, V_t^{\varepsilon_{1:d}}, W_t^{\varepsilon_{1:d}})$.

By Lemma 1, we have, for all $\varepsilon_1, \ldots, \varepsilon_d \in [-1,1], i \in [d], j \in [n]$, and $p \in [0,1]$,

$$\mathbb{E}\left[\mathrm{GFT}_{j+(i-1)n}^{\varepsilon_{1:d}}(p)\right] = 2\int_0^p \int_0^\lambda f_{\varepsilon_i}(s)\,\mathrm{d}s\,\mathrm{d}\lambda + 2(\bar{\nu}_{\varepsilon_i} - p)\int_0^p f_{\varepsilon_i}(s)\,\mathrm{d}s\,,$$

which, together with the fundamental theorem of calculus —(Bass, 2013, Theorem 14.16), noting that $p \mapsto \mathbb{E}\left[\mathrm{GFT}_{j+(i-1)n}^{\varepsilon_{1:d}}(p)\right]$ is absolutely continuous with derivative defined a.e. by $p \mapsto 2(\bar{\nu}_{\varepsilon_i} - p)f_{\varepsilon_i}(p)$— yields, for any $p \in J_L$,

$$\mathbb{E}\left[\mathrm{GFT}_{j+(i-1)n}^{\varepsilon_{1:d}}(\bar{\nu}_{\varepsilon_i})\right] - \mathbb{E}\left[\mathrm{GFT}_{j+(i-1)n}^{\varepsilon_{1:d}}(p)\right] = L|\bar{\nu}_{\varepsilon_i} - p|^2\,. \tag{5}$$

Note also that for all $\varepsilon_1, \ldots, \varepsilon_d \in [-\bar{\varepsilon}_L, \bar{\varepsilon}_L], t \in [T]$, and $p \in [0,1] \smallsetminus J_L$, a direct verification shows that

$$\mathbb{E}\left[\mathrm{GFT}_t^{\varepsilon_{1:d}}(p)\right] \le \mathbb{E}\left[\mathrm{GFT}_t^{\varepsilon_{1:d}}(1/2)\right]\,. \tag{6}$$

Fix any arbitrary deterministic algorithm for the full feedback setting $(\alpha_t)_{t\in[T]}$, i.e., (given that the contexts $c_1, \ldots, c_T$ are here fixed and declared ahead of time to the learner), a sequence of functions $\alpha_t \colon ([0,1] \times [0,1])^{t-1} \to [0,1]$ mapping past feedback into prices (with the convention that $\alpha_1$ is just a number in $[0,1]$). For each $t \in [T]$, define $\tilde{\alpha}_t \colon ([0,1] \times [0,1])^{t-1} \to J_L$ equal to $\alpha_t$ whenever $\alpha_t$ takes values in $J_L$, and equal to $1/2$ otherwise. Define $Z_1 \coloneqq \frac{1+E_1}{2}, \ldots, Z_d \coloneqq \frac{1+E_d}{2}$.

Now, note the following

$$\sup_{\varepsilon_{1:d}\in[-\bar{\varepsilon}_L, \bar{\varepsilon}_L]^d} \sum_{i=1}^d \sum_{j=1}^n \mathbb{E}\left[\mathrm{GFT}_{j+(i-1)n}^{\varepsilon_{1:d}}(\bar{\nu}_{\varepsilon_i}) - \mathrm{GFT}_{j+(i-1)n}^{\varepsilon_{1:d}}\left(\alpha_t(V_1^{\varepsilon_{1:d}}, W_1^{\varepsilon_{1:d}}, \ldots, V_{j-1+(i-1)n}^{\varepsilon_{1:d}}, W_{j-1+(i-1)n}^{\varepsilon_{1:d}})\right)\right]$$

$$\overset{(6)}{\ge} \sup_{\varepsilon_{1:d}\in[-\bar{\varepsilon}_L, \bar{\varepsilon}_L]^d} \sum_{i=1}^d \sum_{j=1}^n \mathbb{E}\left[\mathrm{GFT}_{j+(i-1)n}^{\varepsilon_{1:d}}(\bar{\nu}_{\varepsilon_i}) - \mathrm{GFT}_{j+(i-1)n}^{\varepsilon_{1:d}}\left(\tilde{\alpha}_t(V_1^{\varepsilon_{1:d}}, W_1^{\varepsilon_{1:d}}, \ldots, V_{j-1+(i-1)n}^{\varepsilon_{1:d}} W_{j-1+(i-1)n}^{\varepsilon_{1:d}})\right)\right]$$

$$\overset{\spadesuit}{=} L \sup_{\varepsilon_{1:d}\in[-\bar{\varepsilon}_L, \bar{\varepsilon}_L]^d} \sum_{i=1}^d \sum_{j=1}^n \mathbb{E}\left[\left|\bar{\nu}_{\varepsilon_i} - \tilde{\alpha}_t(V_1^{\varepsilon_{1:d}}, W_1^{\varepsilon_{1:d}}, \ldots, V_{j-1+(i-1)n}^{\varepsilon_{1:d}}, W_{j-1+(i-1)n}^{\varepsilon_{1:d}})\right|^2\right]$$

$$\ge L \sum_{i=1}^d \sum_{j=1}^n \mathbb{E}\left[\left|\bar{\nu}_{E_i} - \tilde{\alpha}_t(V_1^{E_{1:d}}, W_1^{E_{1:d}}, \ldots, V_{j-1+(i-1)n}^{E_{1:d}}, W_{j-1+(i-1)n}^{E_{1:d}})\right|^2\right]$$

$$\overset{\heartsuit}{\ge} L \sum_{i=1}^d \sum_{j=1}^n \mathbb{E}\left[\left|\bar{\nu}_{E_i} - \mathbb{E}[\bar{\nu}_{E_i} \mid V_1^{E_{1:d}}, W_1^{E_{1:d}}, \ldots, V_{j-1+(i-1)n}^{E_{1:d}}, W_{j-1+(i-1)n}^{E_{1:d}}]\right|^2\right]$$

$$= \frac{L}{196} \sum_{i=1}^d \sum_{j=1}^n \mathbb{E}\left[\left|E_i - \mathbb{E}[E_i \mid V_1^{E_{1:d}}, W_1^{E_{1:d}} \ldots, V_{j-1+(i-1)n}^{E_{1:d}}, W_{j-1+(i-1)n}^{E_{1:d}}]\right|^2\right]$$

$$\overset{\blacklozenge}{\ge} \frac{L}{196} \sum_{i=1}^d \sum_{j=1}^n \mathbb{E}\left[\left|E_i - \mathbb{E}[E_i \mid B_{\frac{1+E_i}{2}, 1+2(i-1)n}, \ldots, B_{\frac{1+E_i}{2}, 2(j-1)+2(i-1)n}]\right|^2\right]$$

$$\overset{\clubsuit}{=} \frac{L}{196} \sum_{i=1}^d \sum_{j=1}^n \mathbb{E}\left[\left|E_i - \mathbb{E}[E_i \mid B_{\frac{1+E_i}{2}, 1}, \ldots, B_{\frac{1+E_i}{2}, 2(j-1)}]\right|^2\right]$$

$$= \frac{L}{49} \sum_{i=1}^{d} \sum_{j=1}^{n} \mathbb{E}\Big[\big|Z_i - \mathbb{E}[Z_i \mid B_{Z_i,1}, \dots, B_{Z_i,2(j-1)}]\big|^2\Big]$$

where ♠ follows from (5) and the fact that $\tilde{\alpha}_t$ takes values in $J_L$; ♥ from the fact that the minimizer of the $L^2(\mathbb{P})$-distance from $\bar{\nu}_{E_i}$ in $\sigma(V_1^{E_{1:d}}, W_1^{E_{1:d}}, \dots, V_{j-1+(i-1)n}^{E_{1:d}}, W_{j-1+(i-1)n}^{E_{1:d}})$ is $\mathbb{E}\big[\bar{\nu}_{E_i} \mid V_1^{E_{1:d}}, W_1^{E_{1:d}}, \dots, V_{j-1+(i-1)n}^{E_{1:d}}, W_{j-1+(i-1)n}^{E_{1:d}}\big]$ (see, e.g., (Williams, 1991, Section 9.4)); ♦ follows from the fact that, by Equation (4) and the independence of $E_i$ from $\big((B_{q,t})_{t\in\mathbb{N},q\in[0,1]}, (\tilde{B}_t)_{t\in\mathbb{N}}, (U_t)_{t\in\mathbb{N}}\big)$, the conditional expectation $\mathbb{E}\big[E_i \mid V_1^{E_{1:d}}, W_1^{E_{1:d}}, \dots, V_{j-1+(i-1)n}^{E_{1:d}}, W_{j-1+(i-1)n}^{E_{1:d}}\big]$ is a measurable function of $B_{\frac{1+E_i}{2},1+2(i-1)n}, \dots, B_{\frac{1+E_i}{2},2(j-1)+2(i-1)n}$, together with the same observation made in ♥ about the minimization of $L^2(\mathbb{P})$ distance; and ♣ follows from the fact that the sequence $\big(B_{\frac{1+E_i}{2},t}\big)_{t\in\mathbb{N}}$ is i.i.d..

Finally, the general term of this last sum is the expected squared distance between the random parameter (drawn uniformly over $[(1-\bar{\varepsilon}_L)/2, (1+\bar{\varepsilon}_L)/2])$ of an i.i.d. sequence of Bernoulli random variables and the conditional expectation of this random parameter given $2(j-1)$ independent realizations of these Bernoullis. A probabilistic argument shows that there exist two universal constants $\tilde{a}, \tilde{b} > 0$ such that, for all $j \geq \tilde{b}L^4$ and each $i \in [d]$,

$$\mathbb{E}\Big[\big|Z_i - \mathbb{E}[Z_i \mid B_{Z_i,1}, \dots, B_{Z_i,2(j-1)}]\big|^2\Big] \geq \tilde{a}\frac{1}{j-1} . \tag{7}$$

At a high level, this is because, in an event of probability $\Omega(1)$, if $j$ is large enough, the conditional expectation $\mathbb{E}[Z_i \mid B_{Z_i,1}, \dots, B_{Z_i,2(j-1)}]$ is very close to the empirical average $\frac{1}{2(j-1)} \sum_{s=1}^{2(j-1)} B_{Z_i,s}$, whose expected squared distance from $Z$ is $\Omega\big(1/(j-1)\big)$. For a formal proof of (7) with explicit constants, we refer the reader to Bolić et al. (2024, Appendix B of the extended arxiv version). Summing over $i \in [d]$ and $j \in [n]$, we obtain that there exist $\varepsilon_1, \dots, \varepsilon_d \in [-1,1]^d$ such that

$$\sum_{i=1}^{d} \sum_{j=1}^{n} \mathbb{E}\Big[\mathrm{GFT}_{j+(i-1)n}^{\varepsilon_{1:d}}(\bar{\nu}_{\varepsilon_i}) - \mathrm{GFT}_{j+(i-1)n}^{\varepsilon_{1:d}}\big(\tilde{\alpha}_t(V_1^{\varepsilon_{1:d}}, W_1^{\varepsilon_{1:d}}, \dots, V_{j-1+(i-1)n}^{\varepsilon_{1:d}}, W_{j-1+(i-1)n}^{\varepsilon_{1:d}})\big)\Big]$$

$$= \Omega(Ld \ln n) = \Omega(Ld \ln T) .$$

# B   PROOF OF THEOREM 4

Fix $L \geq 2$ and $T \in \mathbb{N}$. We will use the very same notation as in the proof of Theorem 2. In particular, the contexts $c_1, \dots, c_T$ are again the same as before and declared ahead of time to the learner. We will show that for each algorithm for contextual brokerage with 2-bit feedback and each time horizon $T$, if $R_T^{\varepsilon_{1:d}}$ is the regret of the algorithm at time horizon $T$ when the traders' valuations are $V_1^{\varepsilon_{1:d}}, W_1^{\varepsilon_{1:d}}, \dots, V_T^{\varepsilon_{1:d}}, W_T^{\varepsilon_{1:d}}$, then $\max_{\sigma_{1:d}\in\{-1,1\}^d} R_T^{(\sigma_1\varepsilon,\dots,\sigma_d\varepsilon)} = \Omega\big(\sqrt{dLT}\big)$ if $\varepsilon = \Theta\big((LT/d)^{-1/4}\big)$ and $T = \Omega(dL^3)$.

Note that for all $\varepsilon_{1:d} \in [-1,1]^d$, $i \in [d]$, $j \in [n]$, and $p < \frac{1}{2}$, if $\varepsilon_i > 0$, then, a direct verification shows that

$$\mathbb{E}\Big[\mathrm{GFT}_{j+(i-1)n}^{\varepsilon_{1:d}}(1/2)\Big] \geq \mathbb{E}\big[\mathrm{GFT}_{j+(i-1)n}^{\varepsilon_{1:d}}(p)\big] . \tag{8}$$

Similarly, for all $\varepsilon_{1:d} \in [-1,1]^d$, $i \in [d]$, $j \in [n]$, and $p > \frac{1}{2}$, if $\varepsilon_i < 0$, then

$$\mathbb{E}\Big[\mathrm{GFT}_{j+(i-1)n}^{\varepsilon_{1:d}}(1/2)\Big] \geq \mathbb{E}\big[\mathrm{GFT}_{j+(i-1)n}^{\varepsilon_{1:d}}(p)\big] . \tag{9}$$

Furthermore, a direct verification shows that, for each $\varepsilon_{1:d} \in [-1,1]^d$ and $t \in [T]$,

$$\max_{p\in[0,1]} \mathbb{E}\big[\mathrm{GFT}_t^{\varepsilon_{1:d}}(p)\big] - \max_{p\in[\frac{1}{7},\frac{2}{7}]} \mathbb{E}\big[\mathrm{GFT}_t^{\varepsilon_{1:d}}(p)\big] \geq \frac{1}{50} = \Omega(1) . \tag{10}$$

Now, assume that $T \geq dL^3/14^4$ so that, defining $\varepsilon := (LT/d)^{-1/4}$, we have that for any $\sigma_{1:d} \in \{-1,1\}^d$, any $i \in [d]$ and any $j \in [n]$, the maximizer of the expected gain from trade

$p \mapsto \mathbb{E}\big[\mathrm{GFT}_{j+(i-1)n}^{(\sigma_1\varepsilon,\ldots,\sigma_d\varepsilon)}(p)\big]$ is at $\frac{1}{2} + \frac{\sigma_i\varepsilon}{196}$ and hence belongs to the spike region $J_L$. If $\sigma_i = 1$ (resp., $\sigma_i = -1$) case, the optimal price for the rounds $1 + (i-1)n, \ldots, in$ belongs to the region $\left(\frac{1}{2}, \frac{1}{2} + \frac{1}{14L}\right]$ (resp., $\left[\frac{1}{2} - \frac{1}{14L}, \frac{1}{2}\right)$). By posting prices in the wrong region $\left[0, \frac{1}{2}\right]$ (resp., $\left[\frac{1}{2}, 1\right]$) in the $\sigma_i = 1$ (resp., $\sigma_i = -1$) case, the learner incurs a $\Omega(L\varepsilon^2) = \Omega\big(\sqrt{L/dT}\big)$ instantaneous regret by (5) and (8) (resp., (5) and (9)). Then, in order to attempt suffering less than $\Omega\big(\sqrt{L/T} \cdot n\big) = \Omega\big(\sqrt{LT/d}\big)$ regret in the rounds $1 + (i-1)n, \ldots, in$, the algorithm would have to detect the sign of $\sigma_i$ and play accordingly. We will show now that even this strategy will not improve the regret of the algorithm (by more than a constant) because of the cost of determining the sign of $\sigma_i$ with the available feedback. Since for any $i \in [d]$ and $j \in [n]$, the feedback received from the two traders at time $j + (i-1)n$ by posting a price $p$ is $\mathbb{I}\{p \le V_{j+(i-1)n}^{(\sigma_1\varepsilon,\ldots,\sigma_d\varepsilon)}\}$ and $\mathbb{I}\{p \le W_{j+(i-1)n}^{(\sigma_1\varepsilon,\ldots,\sigma_d\varepsilon)}\}$, the only way to obtain information about (the sign of) $\sigma_i$ is to post in the costly ($\Omega(1)$-instantaneous regret by Equation (10)) sub-optimal region $\left[\frac{1}{7}, \frac{2}{7}\right]$. However, posting prices in the region $\left[\frac{1}{7}, \frac{2}{7}\right]$ at time $j + (i-1)n$ can't give more information about $\sigma_i$ than the information carried by $V_{j+(i-1)n}^{(\sigma_1\varepsilon,\ldots,\sigma_d\varepsilon)}$ and $W_{j+(i-1)n}^{(\sigma_1\varepsilon,\ldots,\sigma_d\varepsilon)}$, which, in turn, can't give more information about $\sigma_i$ than the information carried by the two Bernoullis $B_{\frac{1+\sigma_i\varepsilon}{2}, 2(j+(i-1)n)-1}$ and $B_{\frac{1+\sigma_i\varepsilon}{2}, 2(j+(i-1)n)}$. Since only during rounds $1 + (i-1)n, \ldots, in$ is possible to extract information about the sign of $\sigma_i$ and, (via an information-theoretic argument) in order to distinguish the sign of $\sigma_i$ having access to i.i.d. Bernoulli random variables of parameter $\frac{1+\sigma_i\varepsilon}{2}$ requires $\Omega(1/\varepsilon^2) = \Omega\big(\sqrt{LT/d}\big)$ samples, we are forced to post at least $\Omega\big(\sqrt{LT/d}\big)$ prices in the costly region $\left[\frac{1}{7}, \frac{2}{7}\right]$ during the rounds $1+(i-1)n, \ldots, in$ suffering a regret of $\Omega\big(\sqrt{LT/d}\big) \cdot \Omega(1) = \Omega\big(\sqrt{LT/d}\big)$. Putting everything together, no matter what the strategy, each algorithm will pay at least $\Omega\big(\sqrt{LT/d}\big)$ regret in each epoch $1 + (i-1)n, \ldots, in$ for every $i \in [d]$, resulting in an overall regret of $\Omega\big(\sqrt{LT/d}\big) \cdot d = \Omega\big(\sqrt{dLT}\big)$.

## C  Proof of Theorem 5

Assume that $d \ge 2$ (for the case $d = 1$, the following proof can be adapted straightforwardly by defining $\phi = 1$ and $c_t = 1/2 + \varepsilon_t$, where $\varepsilon_t$ is an arbitrary small sequence of biases). Let $(a_t)_{t\in\mathbb{N}}$ be a sequence of distinct elements in $[0,1]$ and, for all $t \in \mathbb{N}$, let $c_t := (a_t, 1 - a_t, 0, 0, \ldots, 0)$. Notice that $(c_t)_{t\in\mathbb{N}}$ is a sequence of distinct elements in $[0,1]^2$. Define $\phi := (1/2, 1/2, 0, 0, \ldots, 0)$. Notice that for each $t \in \mathbb{N}$ it holds that $c_t^\top \phi = 1/2$. Let $\varepsilon \in (0, 1/16)$. For any $\theta \in \{0, 1\}$, consider the following probability distribution

$$\mu_\theta := \left(\frac{1}{4} + (1-2\theta)\varepsilon\right)\delta_{-\frac{1}{2}} + \frac{1}{2}\delta_{2(1-\theta)\varepsilon - 2\theta\varepsilon} + \left(\frac{1}{4} - (1-2\theta)\varepsilon\right)\delta_{\frac{1}{2}},$$

where for any $a \in \mathbb{R}$, $\delta_a$ is the Dirac's delta probability distribution centered in $a$. Consider an independent family of random variables $(\xi_{t,\theta}, \zeta_{t,\theta})_{t\in\mathbb{N},\theta\in\{0,1\}}$ such that for any $t \in \mathbb{N}$ and any $\theta \in \{0,1\}$, we have that both $\xi_{t,\theta}$ and $\zeta_{t,\theta}$ are random variables with common distribution $\mu_\theta$. Notice that for each $t \in \mathbb{N}$ and each $\theta \in \{0,1\}$ we have that $\mathbb{E}[\xi_{t,\theta}] = 0 = \mathbb{E}[\zeta_{t,\theta}]$. Define, for each $t \in \mathbb{N}$ and each $\theta \in \{0,1\}$, the random variables $V_{t,\theta} := c_t^\top \phi + \xi_t$ and $W_{t,\theta} := c_t^\top \phi + \zeta_t$. Notice that these are $[0,1]$-valued random variables and that $(V_{t,\theta}, W_{t,\theta})_{t\in\mathbb{N},\theta\in\{0,1\}}$ is an independent family. Now, for each $\theta \in \{0,1\}$ and each $t \in \mathbb{N}$, let

$$p^\#(\theta) \in \underset{p\in[0,1]}{\mathrm{argmax}}\, \mathbb{E}\Big[g\big(p, V_{t,\theta}, W_{t,\theta}\big)\Big],$$

which does exist because the function $[0,1] \to [0,1], p \mapsto \mathbb{E}\Big[g\big(p, V_{t,\theta}, W_{t,\theta}\big)\Big]$ is upper semicontinuous (this can be proved as in Cesa-Bianchi et al. 2024a, Appendix B) and defined on a compact set. Furthermore, note that the previous definition is independent of $t$ because, for any $\theta \in \{0,1\}$, the pairs $(V_{t_1,\theta}, W_{t_1,\theta})$ and $(V_{t_2,\theta}, W_{t_2,\theta})$ share the same distribution for every $t_1, t_2 \in \mathbb{N}$. Fix a learning algorithm for the full-feedback contextual brokerage problem, fix a time horizon $T \in \mathbb{N}$,

and notice that since the contexts $c_1, c_2, \ldots$ are all distinct, it follows that

$$\max_{\theta_1,\ldots,\theta_T \in \{0,1\}^T} \sup_{p^\star:[0,1]^d \to [0,1]} \mathbb{E}\left[\sum_{t=1}^T \Big(g\big(p^\star(c_t), V_{t,\theta_t}, W_{t,\theta_t}\big) - g(P_t, V_{t,\theta_t}, W_{t,\theta_t})\Big)\right]$$

$$= \max_{\theta_1,\ldots,\theta_T \in \{0,1\}^T} \sum_{t=1}^T \left(\sup_{p \in [0,1]} \mathbb{E}\big[g(p, V_{t,\theta_t}, W_{t,\theta_t})\big] - \mathbb{E}\big[g(P_t, V_{t,\theta_t}, W_{t,\theta_t})\big]\right)$$

$$= \max_{\theta_1,\ldots,\theta_T \in \{0,1\}^T} \mathbb{E}\left[\sum_{t=1}^T \Big(g\big(p^\#(\theta_t), V_{t,\theta_t}, W_{t,\theta_t}\big) - g(P_t, V_{t,\theta_t}, W_{t,\theta_t})\Big)\right] =: (\#) .$$

Now, consider an i.i.d. family of Bernoulli random variables $(\Theta_t)_{t \in \mathbb{N}}$ with parameter $1/2$, independent of the whole family $(V_{t,\theta}, W_{t,\theta})_{t \in \mathbb{N}, \theta \in \{0,1\}}$. We have that

$$(\#) \geq \mathbb{E}\left[\sum_{t=1}^T \Big(g\big(p^\#(\Theta_t), V_{t,\Theta_t}, W_{t,\Theta_t}\big) - g(P_t, V_{t,\Theta_t}, W_{t,\Theta_t})\Big)\right]$$

$$= \sum_{t=1}^T \left(\mathbb{E}\Big[g\big(p^\#(\Theta_t), V_{t,\Theta_t}, W_{t,\Theta_t}\big)\Big] - \mathbb{E}\Big[g(P_t, V_{t,\Theta_t}, W_{t,\Theta_t})\Big]\right) =: (\$) .$$

Now, for each $t \in [T]$, we see that

$$\mathbb{E}\Big[g\big(p^\#(\Theta_t), V_{t,\Theta_t}, W_{t,\Theta_t}\big)\Big] = \mathbb{E}\left[\mathbb{E}\Big[g\big(p^\#(\Theta_t), V_{t,\Theta_t}, W_{t,\Theta_t}\big) \mid \Theta_t\Big]\right]$$

$$= \mathbb{E}\left[\max_{p \in [0,1]} \mathbb{E}\Big[g\big(p, V_{t,\Theta_t}, W_{t,\Theta_t}\big) \mid \Theta_t\Big]\right]$$

and long but straightforward computations show that, for each $p \in [0,1]$, it holds that

$$\mathbb{E}\Big[g\big(p, V_{t,\Theta_t}, W_{t,\Theta_t}\big) \mid \Theta_t\Big] = \begin{cases} \frac{1}{4} + \varepsilon(1 - 2\Theta_t) & \text{if } 0 \leq p < \frac{1}{2} - 2\Theta_t\varepsilon + 2(1 - \Theta_t)\varepsilon , \\ \frac{3}{8} + 2\varepsilon^2 & \text{if } p = \frac{1}{2} - 2\Theta_t\varepsilon + 2(1 - \Theta_t)\varepsilon , \\ \frac{1}{4} - \varepsilon(1 - 2\Theta_t) & \text{if } \frac{1}{2} - 2\Theta_t\varepsilon + 2(1 - \Theta_t)\varepsilon < p \leq 1 , \end{cases}$$

from which it follows that

$$\max_{p \in [0,1]} \mathbb{E}\Big[g\big(p, V_{t,\Theta_t}, W_{t,\Theta_t}\big) \mid \Theta_t\Big] = \frac{3}{8} + 2\varepsilon^2 .$$

On the other hand, for each $t \in [T]$, leveraging the freezing lemma (Cesari & Colomboni, 2021, Lemma 8), we have that

$$\mathbb{E}\big[g(P_t, V_{t,\Theta_t}, W_{t,\Theta_t})\big] = \mathbb{E}\Big[\mathbb{E}\big[g(P_t, V_{t,\Theta_t}, W_{t,\Theta_t}) \mid P_t\big]\Big] = \mathbb{E}\left[\Big[\mathbb{E}\big[g(p, V_{t,\Theta_t}, W_{t,\Theta_t})\big]\Big]_{p=P_t}\right]$$

$$= \mathbb{E}\left[\left[\frac{1}{2}\mathbb{E}\big[g(p, V_{t,\Theta_t}, W_{t,\Theta_t}) \mid \Theta_t = 0\big] + \frac{1}{2}\mathbb{E}\big[g(p, V_{t,\Theta_t}, W_{t,\Theta_t}) \mid \Theta_t = 1\big]\right]_{p=P_t}\right]$$

and again, tedious but straightforward computations show that, for each $p \in [0,1]$, it holds that

$$\frac{1}{2}\mathbb{E}\big[g(p, V_{t,\Theta_t}, W_{t,\Theta_t}) \mid \Theta_t = 0\big] + \frac{1}{2}\mathbb{E}\big[g(p, V_{t,\Theta_t}, W_{t,\Theta_t}) \mid \Theta_t = 1\big]$$

$$= \frac{1}{4}\left(\mathbb{I}\left\{p < \frac{1}{2} - 2\varepsilon\right\} + \mathbb{I}\left\{\frac{1}{2} + 2\varepsilon < p\right\}\right) + \left(\frac{5}{16} + \frac{\varepsilon}{2} + \varepsilon^2\right)\left(\mathbb{I}\left\{p = \frac{1}{2} - 2\varepsilon\right\} + \mathbb{I}\left\{p = \frac{1}{2} + 2\varepsilon\right\}\right)$$

$$+ \left(\frac{1}{4} + \varepsilon\right)\mathbb{I}\left\{\frac{1}{2} - 2\varepsilon < p < \frac{1}{2} + 2\varepsilon\right\}$$

$$\leq \frac{5}{16} + \frac{\varepsilon}{2} + \varepsilon^2 .$$

We conclude that

$$(\$) \geq \frac{T}{16} + \left(\varepsilon^2 - \frac{\varepsilon}{2}\right)T ,$$

from which it follows that there exists $\theta_1, \ldots, \theta_T \in \{0,1\}$ such that

$$\sup_{p^\star:[0,1]^d \to [0,1]} \mathbb{E}\left[\sum_{t=1}^T \Big(g\big(p^\star(c_t), V_{t,\theta_t}, W_{t,\theta_t}\big) - g(P_t, V_{t,\theta_t}, W_{t,\theta_t})\Big)\right] \geq \frac{T}{16} + \left(\varepsilon^2 - \frac{\varepsilon}{2}\right)T \geq \frac{T}{32} .$$

