# OpenReview forum: "A Contextual Online Learning Theory of Brokerage"
_ICLR.cc/2025/Conference — ICLR 2025 Conference Withdrawn Submission_

### Official Review · Reviewer_cE4e · 2024-10-30

**Soundness:** 3
**Presentation:** 2
**Contribution:** 2
**Rating:** 5
**Confidence:** 4

**Summary:**

This paper studies an online model of OTC markets, where traders arrive at each round with private valuations, a brokers proposes a price, and traders engage if the price is consistent with their valuations. At each round/product, they assume an unknown ground truth market price $m_t$ exists, with both the traders valuations being sampled from a distribution that is an unbiased estimator of this price. Further, the broker observes a context $c_t$, which is linearly consistent with $m_t$ as follows: $\langle c_t, \phi \rangle = m_t$. The work studies two feedback model (at round's end, either observe the private valuation or just the traders participation) and provide tight bounds in both settings. While I think the technical work here is sound and parts of the results interesting, I am not fully convinced by the conceptual model the problem attempts to study and the additional insights it offers beyond the current literature.

**Strengths:**

- The paper is fairly well-written and key ideas and assumptions exposited quite nicely
- The technical results look sound. The full feedback algorithm based on online ridge regression is reasonable and I am especially intrigued by the algorithm for the 2-bit feedback case, where it essentially attempts to reconstruct the CDF. I would actually appreciate a bit more discussion on the intuition behind the algorithm. Although some of the techniques borrow from previous works in terms of novelty, I don't see this as a shortcoming.
- The technical bounds provided here are nearly tight. Both the full-feedback and 2-bit feedback models have matching lower bounds, and the authors show the necessity of the bounded distributional assumption.

**Weaknesses:**

My major critique of the paper arises from it's conceptual model of the OTC marketplace and the additional insights it provides compared to existing literature.

- In "A Regret Analysis of Bilateral Trade" (Cesa-Bianchi, 2021) at each time step a (buyer, seller) pair arrives, with the buyer having maximum valuation $b_t$ and seller having minimum valuation $s_t$. It is clear that a trade happens if the proposed price $p_t \in [s_t, b_t]$. In this paper, however, the role of the buyer and seller are not determined. Two parties arrive, and depending on the p_t given, either could be a buyer or seller. This is quite strange and only really makes sense if both traders own the underlying asset - so either can be the seller. I am not sure how to justify this/be on board with this model and realistic this is in OTC markets.

- In this model, both trader's valuation arise from a stochastic process, which in expectation, is equal to the market price, which is crucially consistent with the context observed by the learner. So $E[V_t] = E[W_t] = m_t = \langle c_t, \phi \rangle$. This is also quite a strange and strong assumption; this notion of an expected ground truth market price agreed both buyer and seller parties is not present in (Cesa-Bianchi, 2021). In general, market price arises endogenously due to trading between different parties with possibly heterogenous expected valuations. Am I correct to say that if both traders are highly concentrated (low variance), then the GFT should be very small since they both have the same EV? Moreover, it requires consistency with the learner's context. On theoretical models, I am happy to be lenient with assumptions, but I worry that in this case, it does not provide any more natural insights or technical benefits over what is currently proposed in the literature (see below):

- I am trying to understand the novelty or additional insight this paper provides over "An Online Learning Theory of Brokerage" (Bolic, 2024), which the authors cite. That paper does not make a contextual assumption, nor do they make an assumption about mean of the trader's distribution - however, they assume both traders valuations are from same fixed distribution. In this model, trader's don't need to have the same distribution, but the same mean, which is consistent with the market price the context reveals. Beyond these differences, the model is nearly identical and the bounds achieved are also identical. I am not sure if the different distributional assumption can truly be justified as loosening the model since it requires an additional sort of internal consistency with the context - $E[V_t] = E[W_t] = m_t = \langle c_t, \phi \rangle$. I understand why this assumption is made from a technical/learning perspective, but conceptually, I don't see any immediate reasons to think this model is more natural than the one in (Bolic, 2024) or (Cesa Bianchi, 2021) which clearly delineates buyers and sellers, both of which give similar bounds.

- Why use this strange notation for $a \wedge b$ denoting $min(a, b)$ and $a \vee b$ denoting $max(a,b)$? Makes it much harder to parse.

**Questions:**

See the weaknesses mentioned.

---

### Official Review · Reviewer_9Aj8 · 2024-11-03

**Soundness:** 3
**Presentation:** 3
**Contribution:** 2
**Rating:** 6
**Confidence:** 3

**Summary:**

The paper studies a online contextual brokerage problem. The game operates as follows: At each time round $t$:

1. two traders arrive with private valuations $V_t, W_t$ arrives.
2. the broker observes a context $c_t\in\mathbb{R}^d$ and proposes a price $P_t$
3. if the price $P_t$ is between the lowest valuation $V_t\vee W_t$ and highest valuation $V_t\wedge W_t$ (meaning the trader with the minimum valuation is ready to sell at $P_t$ and the trader with the maximum valuation is eager to buy at $P_t$), the asset is bought by the trader with the highest valuation from the trader with the lowest valuation at the brokerage price $P_t$.

The paper assumes that both $V_t, W_t$ are random variables with same expected value $m_t = c_t^\top\phi$ for some unknown vector $\phi$. The the reward of each interaction is the sum of the net utilities of the traders, known as gain from trade. The goal of the learner is to minimize the regret with respect to the best function of the contexts. The paper considers two types feedback: (1) full feedback — both valuations $V_t, W_t$ are revealed to the learner at the end of each round (2) two-bit feedback — only the indicator functions $\mathbb{I}\{P_t\le V_t\}$ and $\mathbb{I}\{P_t\le W_t\}$ are disclosed.

Let $L$ denotes the upper bounded density of the valuation distributions. The paper shows that in the full feedback setting, an ridge regression estimation-based algorithm can achieve a tight $\Theta(Ld\ln T)$ regret. For the two-bid feedback, the paper shows an tight $\Theta(\sqrt{Ld T})$ regret bound. The paper concludes by showing that the bounded density assumption is necessary to obtain subliner regrets.

**Strengths:**

The studied problem is well-motivated and interesting. Previous works approaching bilateral trade problem focus on a context-free setting, while this work introduces context to the problem. Obtaining tight regret bounds are not trivial in this contextual setting. A nice result of this paper is that they also establish the necessities of the bounded density assumption.

**Weaknesses:**

Maybe one concern is that there seems not much novelty in the developed algorithm. For example, for the algorithm of the full-feedback, the algorithm seems to be a direct application of using the regression to compute the estimate of the unknown vector. The authors may consider adding discussions about the novelty of the proposed algorithm, or the challenges of the learner's problem.

**Questions:**

I am wondering if some results could be also obtained if the learner only has one-bit feedback, namely only observe whether the trade happens or not.

---

### Official Review · Reviewer_iUQE · 2024-11-08

**Soundness:** 3
**Presentation:** 3
**Contribution:** 1
**Rating:** 5
**Confidence:** 3

**Summary:**

This paper discusses online bandit learning for bilateral trade and is largely a follow-up to [1]. In this brokerage game, a buyer and seller share equal valuations on average, and it is the strategy of the broker to set the brokerage price, $ p $, as close as possible to this amount, $ m $, to maximize the "gain from trade." A sale only occurs when $ p $ falls between realized values $ V $ and $ W $, which are random variables distributed around $ m $. This repeated online learning setting was investigated in [1], and the current work adds a context that causally affects $ E[m] $ via a set of $ d $ parameters. The paper presents two problem settings identical to [1]: full-feedback, where the realized valuations of the buyer and seller are revealed to the agent, and two-bit feedback, where only whether a sale occurred is revealed, making it more challenging. Two algorithms based on ridge regression are proposed to address this problem, each providing logarithmic regret bounds. Additional theorems establish a lower bound on regret under specific revelation of context sequences, and the final theorem, Theorem 5, addresses the (un)learnability of this problem when a core bounded probability assumption is lifted.

[1] Bolić, Nataša, Tommaso Cesari, and Roberto Colomboni. "An online learning theory of brokerage." arXiv preprint arXiv:2310.12107 (2023).

**Strengths:**

- This paper addresses a very relevant and practical economic problem: brokering a sale between parties under a fair valuation for a product, involving quantified uncertainty.
- The mathematical tools used to obtain the theoretical results are advanced and innovative.
- The proofs are rigorous, and some theorems provide non-trivial results, especially regarding the learnability of the problem under specific circumstances.

An additional note: Theorem 5 stipulates that there will be best-case linear regret with an unbounded $ L $, making a finite $ L > 1 $ a valid requirement. However, from my perspective, the assumption of a finite $ L $ in a bounded interval is quite mild, and it's safe to say that most random distributions in this economic setting would adhere to it.

**Weaknesses:**

- The major weakness is the level of contribution compared to previous work [1] and the amount of verbatim text copied from one document to another. As mentioned, the problem setting is almost identical, except that additional context is provided. A word-for-word comparison between the two documents shows high similarity scores for the first half of the document. If this is a follow-up to [1], it should be stated more clearly, and some redundancy could be reduced by not rewriting the exact same language from [1] in the current draft.
- In reviewing the theoretical results, there seems to be some repetition: Lemma 1 provides the same result as Thm 2.3 in [1], and Theorems 1 and 3 demonstrate almost identical regret bounds as Algorithm 1 and 2 in [1], albeit with different algorithms and arguments. This somewhat reduces the degree of contribution, as adding context to the same problem as [1] results in finding a new algorithm that retains nearly identical regret bounds as [1], which makes the result rather incremental.
- Economic assumptions: The assumption that both the buyer and seller have the exact same valuation seems somewhat unrealistic. Isn’t it generally the case that, in markets, parties value goods differently? Furthermore, why should agents act under a true valuation? Could they not be strategic in their decisions? It also appears that the broker is entirely altruistic—facilitating a trade but seemingly not profiting from it.

**Questions:**

- It seems the broker is completely altruistic. Could brokers not typically profit by imposing a spread between buy and sell prices? So, could $ p $ be a range rather than a single point, and what impact would this have on the current work?
- What would happen if the buyer and seller did not declare their prices truthfully? (This is more of an extension question, but in an open market, this seems like the more common case.)
- Since Lemma 1 reaches the same conclusion as Thm 2.3 in [1], couldn’t we simply use the result from [1]?
- Why must the buyer and seller have the same valuation? Could $ E[V] \neq E[W] $?
- If I understand correctly, Thms. 2 and 4 state that it’s impossible to have sub-logarithmic performance guarantees for a specific sequence of contexts. In what ways is this significant to the paper? And how could it be relevant for future work?

---

### Official Review · Reviewer_WAiA · 2024-11-09

**Soundness:** 3
**Presentation:** 4
**Contribution:** 2
**Rating:** 5
**Confidence:** 4

**Summary:**

This paper analyzes the problem of online learning of bilateral trading in the contextual setting. The authors provide a comprehensive regret analysis for different feedback settings. In particular, the authors analyze two feeback models, full information feedback model and the two-bit feedback and they show tight regret analysis (upper bound and matching lower bounds) for both settings.

**Strengths:**

The paper analyzes a very interesting problem, online bilateral trade. I am not quite convinced the algorithm proposed in this paper will be very useful in practice or can be really applied to any real system, but this is still an interesting theoretical paper and my review is mainly based on this point.

The theoretical analysis is sound and the conveys a complete story.

The paper is well written.

**Weaknesses:**

I am not convinced the paper passes the bar of ICLR.

This paper seems an extension to the previous paper "An online learning theory of brokerage" and generalizes the results to the contextual setting. Can you elaborate more what is the main challenge this paper addresses on top of the previous paper?

Some related work is missing https://arxiv.org/abs/2405.18183, which has been posted to ArXiv this May, that analyzes a very similar setting, where the authors in that paper also discussed single-bit feedback model. If possible, can the authors compare a bit with that one?

**Questions:**

One question regarding the comparison with https://arxiv.org/abs/2405.18183, it seems in their paper, they show a matching lower bound $T^{2/3}$ for the two-bit feedback model, noisy distribution, strong budget balance (propose the same trading price for seller and buyer, which is the same as your setting), however, you achieved $O(\sqrt{T\log T})$ regret. Can you elaborate a bit more here? Is it because the seller and buyer in your setting shares the same expected valuation (market price)?

---

### Author Response · Authors · 2024-11-19
**Thank you for your feedback!**

Dear reviewers, we thank you all for your invaluable comments and suggestions.

The biggest concern, which most of you seem to share, is the technical novelty of this work. In particular, compared to its non-contextual counterpart [1].

To make a long story short, adding context to a setting is a major change that requires algorithmic and proof ideas that are totally orthogonal to those of their non-contextual counterparts. This is because, in the presence of contexts, the problem goes from *learning values* to *learning functions*---a vastly more challenging task.
Also, consider the two-bit feedback setting. In the absence of contexts, it is clear that it is optimal to carry out the entire exploration first and then to stick to exploitation. However, with contexts, the relevance of exploration depends on the context, so we need to balance exploration and exploitation in an online fashion.

That said, the fact that multiple reviewers asked for similar clarifications worried us that our message was not conveyed effectively.

Therefore, we prefer to withdraw the paper, spend some time reworking the writing, incorporating all your precious comments, and resubmit it later.

We sincerely thank you for your hard work and for taking time off of your busy schedule to review our submission.

Warm regards,
The authors

---

### Note · Authors · 2024-11-19

**Comment:**

Dear reviewers, we thank you all for your invaluable comments and suggestions.

The biggest concern, which most of you seem to share, is the technical novelty of this work. In particular, compared to its non-contextual counterpart [1].

To make a long story short, adding context to a setting is a major change that requires algorithmic and proof ideas that are totally orthogonal to those of their non-contextual counterparts. This is because, in the presence of contexts, the problem goes from *learning values* to *learning functions*---a vastly more challenging task. Also, consider the two-bit feedback setting. In the absence of contexts, it is clear that it is optimal to carry out the entire exploration first and then to stick to exploitation. However, with contexts, the relevance of exploration depends on the context, so we need to balance exploration and exploitation in an online fashion.

That said, the fact that multiple reviewers asked for similar clarifications worried us that our message was not conveyed effectively.

Therefore, we prefer to withdraw the paper, spend some time reworking the writing, incorporating all your precious comments, and resubmit it later.

We sincerely thank you for your hard work and for taking time off of your busy schedule to review our submission.

Warm regards,

The authors

**Withdrawal Confirmation:**

I have read and agree with the venue's withdrawal policy on behalf of myself and my co-authors.